# Management of Inadvertent Arterial Catheterization during Central Venous Catheter Placement: A Case Series

**DOI:** 10.3390/jpm12091537

**Published:** 2022-09-19

**Authors:** Georgios Papastratigakis, Diamantina Marouli, Athanasia Proklou, Nikolaos Nasis, Eumorfia Kondili, Elias Kehagias

**Affiliations:** 1Department of Anesthesiology, University Hospital of Heraklion, Voutes, 71110 Heraklion, Greece; 2Department of Intensive Care, University Hospital of Heraklion and Medical School, University of Crete, Voutes, 71110 Heraklion, Greece; 3Interventional Radiology Unit, Department of Medical Imaging, University Hospital of Heraklion and Medical School, University of Crete, Voutes, 71110 Heraklion, Greece

**Keywords:** inadvertent arterial catheterization, central venous catheter, ultrasound guidance

## Abstract

Percutaneous central venous catheterization, although a widely used technique in ICU patients worldwide, is not devoid of complications even under real-time ultrasound guidance. Arterial puncture is a well-recognized complication, while unintended subclavian or carotid artery cannulations during attempted central venous catheterization are infrequent, but documented complications with potentially deleterious consequences. Recently, endovascular balloon tamponade has emerged as the preferred initial approach to repair inadvertent arterial cannulations. Herein, we present a case series of inadvertent arterial catheterization during an attempted ultrasound-guided access of the right internal jugular and the left subclavian vein that were successfully managed with endovascular balloon tamponade.

## 1. Introduction

Central venous catheters (CVCs) are essential for the optimal management and safe delivery of drugs in ICU patients worldwide. More than 5 million CVCs are inserted in the United States each year [1]. However, central venous catheterization is not devoid of complications even under real-time ultrasound guidance [2]. Complications during CVC placement have been classified as either infectious, thrombotic, or mechanical (i.e., related to the procedure of CVC placement itself). The frequency of mechanical complications varies according to the site of CVC insertion, physician experience, and the use of ultrasound guidance. Literature suggests that internal jugular vein catheterization and subclavian vein catheterization carry similar risks of mechanical complications (6.3–11.8% vs. 6.2–10.7%, respectively), whereas femoral vein catheterization carries the highest risk of mechanical complications (12.8–19.4%) and therefore is generally not recommended unless other insertion sites are contraindicated [1]. The most common mechanical complications during CVC insertion in the internal jugular vein include arterial puncture, hematoma (<0.1–2.2%), and pneumothorax (<0.1–0.2%) [1]. Arterial puncture is a well-recognized complication that occurs in 6.3–9.4% of internal jugular vein cannulation attempts [1] and it is by far the most commonly encountered mechanical complication [1]. Carotid artery cannulation by a large bore catheter is a rare complication occurring in 0.1% to 0.5% of cases [3]. The remaining percentage refers to cases where the arterial puncture was noticed in time, and proper measures were taken, and as such, no accidental carotid cannulation occurred.

Accidental subclavian artery catheterization during attempted catheterization of the internal jugular vein is an even rarer, but documented, complication [1,3,4,5,6,7,8], with possible deleterious consequences. We present a case series of patients that underwent an inadvertent arterial catheterization during an attempted ultrasound-guided access of the right internal jugular and the left subclavian vein.

## 2. Case 1

A 40-year-old female patient was admitted to the multidisciplinary ICU of the University Hospital of Heraklion due to a wake-up stroke. Subsequent brain CT revealed a left-sided middle cerebral artery thrombotic stroke, a left cavernous carotid artery calcification, a midline shift, and some hemorrhagic elements in the left temporal lobe. Due to neurologic deterioration the patient underwent emergency endotracheal intubation, and following neurosurgical consultation, an emergency left-sided decompressive craniectomy was performed. Postoperatively, the patient was admitted to the ICU for further management.

On the 11th postoperative day, new onset fever (up to 39.2 °C) accompanied by hemodynamic instability necessitated CVC removal according to existing guidelines and our ICU’s protocol.

The right internal jugular vein was selected for a new CVC insertion. The patient was placed in the Trendelenburg position. After skin prepping, landmark identification, and local anesthetic site infiltration, an ultrasound probe was used to identify the internal jugular vein for introducer needle insertion. The introducer needle was inserted under real-time ultrasound guidance with negative pressure, until a dark-colored nonpulsatile blood was aspirated, and then the needle has held firmly into place. The aspirated blood was sent for blood gas analysis, which confirmed its venous origin (pH: 7.31, pO2: 38 mmHg, pCO2: 49 mmHg, ScvO2: 69%). The guidewire was advanced through the introducer needle and the dilator and central venous catheter were subsequently advanced over the guidewire, with no apparent resistance. All catheter ports were flushed with saline. There was spontaneous filling of the three lumens with blood and visible arterial pulsations. A new blood sample was drawn and sent for blood gas analysis, which indicated that the catheter had been placed intra-arterially (pH: 7.37, pO2: 86 mmHg, pCO2: 39 mmHg, SpO2: 99%). Pressure waveform obtained by catheter transducer confirmed the intra-arterial placement. The anatomical proximity of the carotid artery initially made it the most probable vessel to have been catheterized. The catheter was left in situ and sutured into place.

Emergency CT angiography confirmed that the catheter had transversed the internal jugular vein and had been inserted into the right subclavian artery. Ultrasound Doppler examination of the right upper extremity confirmed presence of a biphasic signal in the brachial radial and ulnar arteries. Following vascular surgical and interventional radiology consultation, intravascular balloon tamponade with concurrent external pressure application was decided as the preferred method for safe catheter removal. The patient was transported to the Interventional Radiology Unit, (Figure 1A) where an 8 × 40 mm Mustang angioplasty balloon (Boston Scientific, Marlborough, MA, USA) was inserted through the right brachial artery, which was immediately inflated after the CVC removal, providing internal balloon tamponade while simultaneous external pressure was applied to the insertion site for 5 min. Subsequent angiography revealed no leaks or fistulae (Figure 1B). No further complications from the accidental intra-arterial catheter insertion were noted. On the 18th day of ICU stay, the patient was discharged to the Neurology Department, with a GCS of 10/15 (Eye: 4/4, Motor 5/6, Verbal: 1/5) and neurological findings similar to those noted on hospital admission.

## 3. Case 2

An 86-year-old male patient was admitted to the multidisciplinary ICU department of the University General Hospital of Heraklion due to type I respiratory insufficiency.

The right internal jugular vein was selected for CVC insertion and the patient was placed in the Trendelenburg position. After skin prepping, ultrasound-guided access of the right IJV was performed, using the out-of-plane technique. The guidewire was advanced through the introducer needle and the dilator and central venous catheter were subsequently advanced over the guidewire. No resistance was felt during the procedure. During saline flushing, pulsatile blood was observed spontaneously filling the ports. Transducement of the three-lumen catheter confirmed the arterial placement. The catheter was left in situ and was sutured in place. A central venous catheter was subsequently placed in the patient’s right femoral vein. No abnormal neurological findings were reported at the time. Due to the patient’s deteriorating respiratory insufficiency, he was intubated.

After initial patient stabilization, emergency CT angiography (Figure 2A) confirmed that the catheter had been inserted into the right common carotid artery. Vascular surgeons and interventional radiologists were consulted and external pressure application was decided as the preferred method for safe catheter removal. Alternatively, upon plan failure, internal intravascular balloon tamponade was to be used concurrently with external pressure application. The patient was transported to the Interventional Radiology Unit (Figure 2B), where a 5F sheath and catheter were inserted through the right brachial artery. The patient reported bradycardia at approximately 45–50 bpm that was noted upon patient admission to the ICU and an isoproterenol infusion was initiated and titrated to approximately 65 bpm. The patient was additionally monitored with bilateral cerebral oximetry monitoring. The catheter was slowly removed and firm external pressure was applied to the site of entry for 15 min. No significant changes were observed to the patient’s cerebral oximetry throughout the procedure. Subsequent angiography revealed no leaks or fistulae (Figure 2C). No further complications from the accidental intra-arterial catheter insertion were noted and no abnormal neurological findings were noted upon the patient’s awakening trial on his 4th and 5th day of admission.

## 4. Case 3

A 65-year-old male with lung carcinoma was referred for right central venous port removal due to skin ulcer formation and for port placement on the left. After skin antiseptic preparation and local anesthesia, the left subclavian vein (SCV) was accessed using real-time ultrasound guidance in the in-plane approach. The left SCV was entered using aspiration with a saline filled syringe. No pulsatile blood flow was identified. The guide-wire was introduced and followed on fluoroscopy crossing the diaphragm to the right of the spinal column. After the implantable port’s 8F peel-apart sheath was introduced, it was realized that it actually was intra-arterially. The peel-away sheath was removed and a Perclose proglide 6F arterial closure device was introduced and deployed over the guide-wire. After pulling the sutures of the Perclose hemostasis was achieved, unfortunately the sutures inadvertently broke due to excessive pulling (Figure 3A). Manual compression was immediately applied. The left brachial artery was accessed using local anesthesia, under real time US-guidance, with a 5F sheath. A guide-wire was introduced through the 5F sheath and angiography was performed using a diagnostic catheter to identify the site of the extravasation. A 7 × 40 mm Mustang angioplasty balloon was navigated to the extravasation using fluoroscopy and inflated to nominal pressure for 5 min (Figure 3B). After a repeat angiography, a small extravasation was still visible in the left SCA. The balloon was again inflated for 5 min with concomitant manual compression and at the final run, no extravasation was visible (Figure 3C). A few days later, the right port was removed and a right PICC was inserted for vascular access.

## 5. Discussion

Accidental carotid and subclavian arterial puncture is a well-documented complication that occurs in 6.3–9.4% of internal jugular vein and 3.1–4.9% of subclavian vein central venous catheterization attempts, respectively [1]. Although rare, both subclavian artery injury [4,5,6,7] and accidental subclavian artery catheterization [8,9,10,11,12,13,14] during attempted catheterization of the internal jugular vein are documented complications which are probably under-reported.

In our first and second case, subclavian artery and common carotid were accidentally accessed with a CVC through the right IJV, and this was mainly the reason why insertion of a closure device was not attempted. In addition, in case 2, the CVC insertion point was very close to the origin of the right vertebral artery which could be compromised. In case 3, use of a closure device was attempted but failed. Also in case 3, the fact that the guide-wire happened to cross the diaphragm to the right of the thoracic spine was not pathognomonic of an intravenous position—as it would be for a right-sided access—due to a descending aorta deviation to the right.

The current American Society of Anesthesiologists (ASA) practice guidelines [2] on CVC insertion, heavily emphasize the prevention of mechanical trauma or injury as a means to reduce the overall number of complications. Confirmation of venous placement of a thin-wall needle or of a catheter that went over the needle prior to wire threading is recommended. Appropriate methods include, but are not limited to, the use of ultrasound, manometry, or pressure–waveform analysis measurement. It should be heavily stated that blood color or absence of pulsatile flow is not a sensitive indicator to confirm that the catheter or thin wall needle resides inside the vein. After confirmation of venous placement, wire threading is allowed to take place. If there is any uncertainty that the catheter or wire resides inside the vein, confirmation is required using ultrasound, TEE, continuous ECG, or fluoroscopy. After venous residence of the wire has been confirmed, insertion of a dilator or large bore catheter may then proceed [2].

After catheterization and before use, catheter residence in the venous system should again be confirmed as soon as clinically appropriate via either manometry, pressure-waveform measurement, or contrast-enhanced ultrasound [2]. In our first and second cases, the decision to confirm the suspected intra-arterial placement of the catheter was hastened by the fact that spontaneous filling of the three lumens with pulsating blood was observed, a finding that suggests intra-arterial placement being probable. Intra-arterial placement of the CVC was confirmed in both cases by the use of the pressure–waveform measurement.

Real-time ultrasound (US) guidance has been an integral part of central venous catheterization guidelines [2], being recommended as a method to improve overall success rates and reduce rates of accidental arterial puncture. However, the clavicle and adjacent bone structures impede the US beam, rendering direct US visualization of subclavian arterial injury and/or catheterization difficult. In our first case, despite real-time US guidance, subclavian artery catheterization was not visualized and was subsequently suspected via clinical features. Immediate confirmation of the iatrogenic injury is of utmost importance to determine the appropriate treatment plan. Leaving the catheter in place and not using the ‘’pull and pressure’’ technique immediately after accidental arterial cannulation is also important, as this technique has been correlated with a number of complications, such as cerebral infarction, pseudoaneurysm, arteriovenous fistulae, and massive hemorrhage that may require urgent open surgical intervention [2,12]. However, complications were much more uncommon in cases where the misplaced catheter was left in place and vascular surgeons and/or interventional radiologists were consulted [2].

In case of a confirmed accidental intra-arterial cannulation, a number of promising therapeutic approaches are available according to the literature. Current ASA guidelines [2] state that, when such an inadvertent cannulation of an arterial vessel occurs, the catheter should be left in place and immediate consultation with a general surgeon, a vascular surgeon, or an interventional radiologist should be sought to construct an appropriate plan for the catheter removal. Treatment should be guided based on existing resources and expertise, and should be individualized based on a number of factors (e.g., patient comorbidities, anticoagulation status, CVC diameter, site of injury, time taken for injury to be recognized, etc.) [2].

Treatment options include open surgical exploration and vascular repair [9,12], percutaneous closure devices [8,13], or, as in our case, temporary balloon tamponade with concurrent external manual pressure application [15] or external manual compression in a palpable artery like the common carotid, where the balloon tamponade is best avoided. No specific guidelines or high-level evidence exist for the proper management of these patients, and decisions are best individualized. However, based on literature reviews, a number of treatment algorithms have been proposed [16,17]. Open surgery seems to be the optimum treatment; however, endovascular treatments seem to be only marginally less successful (94.6% for successful endovascular repair vs. 100% for open repair), and treatment should be individualized, as endovascular treatment offers advantages over open surgery in some cases [17]. Endovascular repair might be an attractive option for patients with significant comorbidities or with acute illness, that might benefit by avoiding general anesthesia. Endovascular techniques should also especially be considered for injuries to the subclavian artery that is not only difficult to compress but also difficult to access surgically [16,17]. Open surgical management is recommended when the injury is recognized more than 4 h after cannulation, or when there is no endovascular treatment service [17].

## Figures and Tables

**Figure 1 jpm-12-01537-f001:**
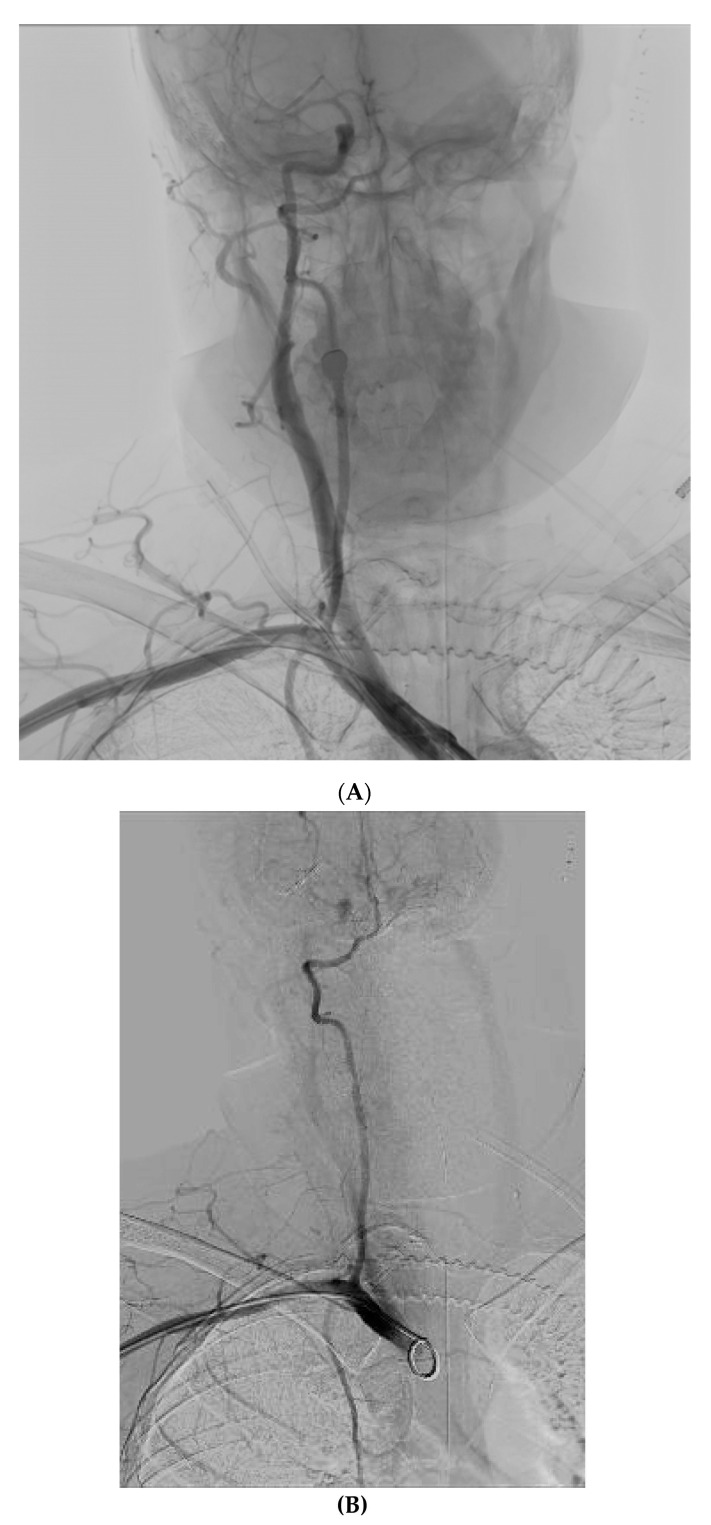
(**A**) CT angiography confirming right subclavian artery catheterization, just distal to the origin of the right vertebral artery. (**B**) Final result after removal of the CVC with no extravasation.

**Figure 2 jpm-12-01537-f002:**
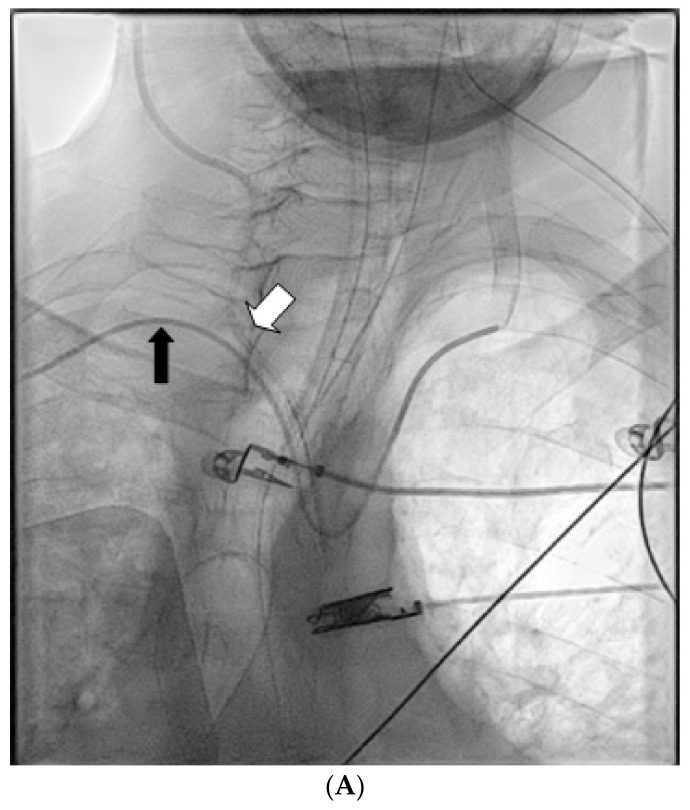
(**A**) Digital angiography confirming right common carotid artery (CCA) catheterization with a CVC (white arrow). An angiography catheter in the arterial system is also visible (black arrow). (**B**) Angiography during removal of the CVC and compression of the CCA. (**C**) Final result after release of compression, 5 min later with no evidence of extravasation.

**Figure 3 jpm-12-01537-f003:**
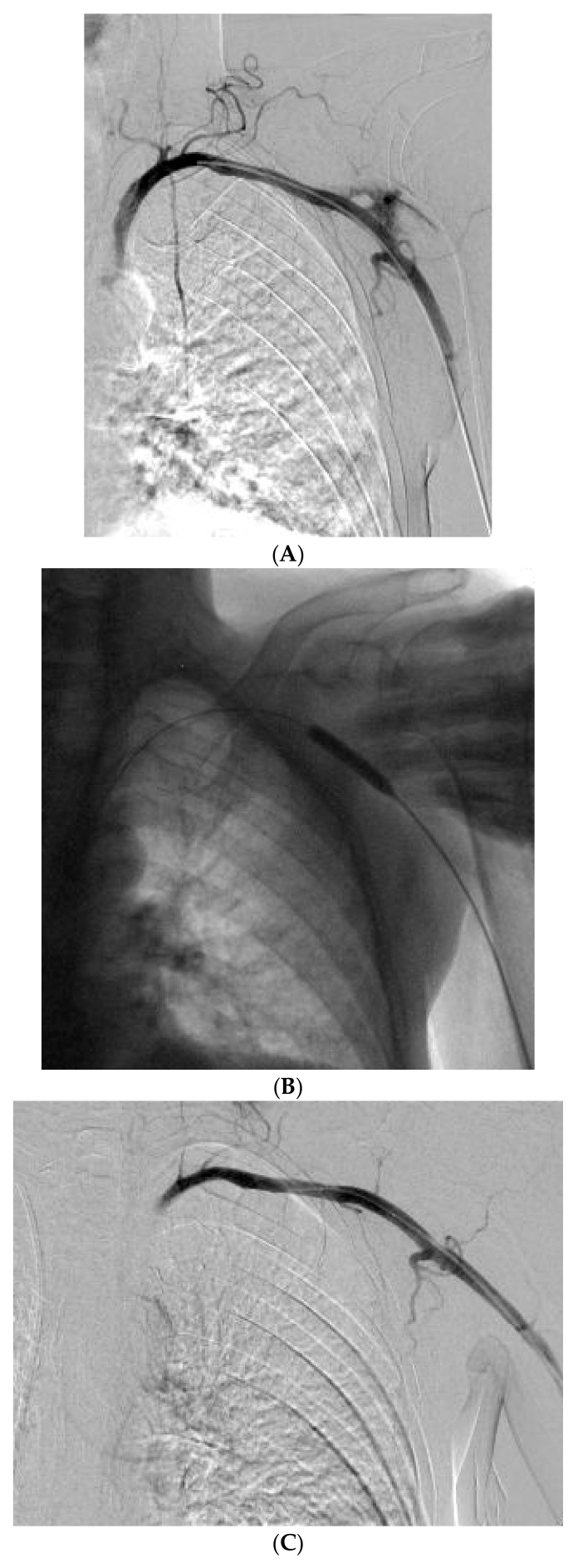
(**A**) Contrast extravasation from the left SCA, after closure device failure. (**B**) Manual compression with balloon assisted inflation at the site of the extravasation. (**C**) Final result, with no extravasation.

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
