# Peer review of "Management of Inadvertent Arterial Catheterization during Central Venous Catheter Placement: A Case Series"

_jpm, 2022, doi:10.3390/jpm12091537_

Round 1

Reviewer 1 Report

Introduction: If arterial cannulation occurs in 6-9% and carotid cannulation is accounting for 0.1-0.5% of the cases what accounts for the remaining?

Author Response

We would like to thanks the reviewer for the valuable comment.

 Comment :Introduction: If arterial cannulation occurs in 6-9% and carotid cannulation is accounting for 0.1-0.5% of the cases what accounts for the remaining?

Response: As stated in our text, Carotid Artery Punctureoccurs in 6.3-9.4% of IJV cannulation cases, and Carotid Artery Cannulation occurs in 0.1-0.5% of IJV cannulation cases. The remaining percentage refers to cases where the arterial puncture was noticed in time, and proper measures were taken, and as such, no accidental carotid cannulation occurred. In the revised manuscript we have add  a sentence in the introduction  to specify this point.

Reviewer 2 Report

Inadvertent arterial catheterization is a rare complication of central venous catheters and is of clinical significance if timely identification and proper interventions are taken. This article reported a series of detailed cases of inadvertent arterial catheterization and outcomes, which is referential to other clinical operators.

I would suggest following possible modifications that may help to delivery the comprehension to readers.

1. Please extend introduction party by providing more gap in knowledge and malignance outcomes.

2. Please re-arrange all the figures.

3. Please provide more results of laboratory tests as supporting information.

4. Please summarize the suggested process for the identification and proper interventions.

In summary, I would recommend acceptance of this article as case report by JMI after minor reversion.

Author Response

 Response to reviewer #2

We are grateful to the reviewer for  the valuable comments which will  help to improve our manuscript

Please find bellow a point to point response to  the reviewer’s comments

 Comment 1: Please extend introduction party by providing more gap in knowledge and malignance outcomes.

Response: We agree with the reviewer  and in revised manuscript we have extended the introduction  and  we have added a new paragraph with  possible complications  as follow:

<< Complications during CVC placement have been classified as either infectious, thrombotic or mechanical (i.e., related to the procedure of CVC placement itself).The frequency of mechanical complications varies according to the site of CVC insertion, physician experience, and the use of ultrasound guidance. Literature suggests that internal jugular veincatheterization and subclavian vein catheterization carry similar risks of mechanical complications (6.3-11.8% vs 6.2-10.7% respectively), whereas femoral vein catheterization carries the highestrisk of mechanical complications (12.8-19.4%) and therefore is generally not recommended unless other insertion sites are contraindicated1.The most common mechanical complications during CVC insertion in the internal jugular vein include arterial puncture, hematoma (<0.1-2.2%) andpneumothorax (<0.1-0.2%)1>>

Comment 2# Please re-arrange all the figures.

Response:  We agree with reviewer and in the revised manuscript we have  re-arranged all the figures

 Comment3# Please provide more results of laboratory tests as supporting information.

Response : According to the reviewer’s suggestion in the revised manuscript we have added the blood gas analysis of both arterial and venous  specimen  for the case 1 .

 Comment 4#. Please summarize the suggested process for the identification and proper interventions.

Response :  We agree with reviewer and according to his/her comment in the revised  manuscript in the discussion section we  provide the suggested process for the identification  (paragraph 3&4) and proper interventions (paragraph 5&6) as follow :

<< Current American Society of Anesthesiologists (ASA) practice guidelines2on CVC insertion, heavily emphasize prevention of mechanical trauma or injury as a means to reduce theoverallnumber of complications. Confirmation of venous placement of a thin-wall needle orof a catheter that went over the needle, prior to wire threading, is recommended. Appropriate methods include but are not limited to the use of ultrasound, manometry or pressure-waveform analysis measurement. It should be heavily stated that blood color or absence of pulsatile flow is not a sensitive indicator to confirm that the catheter or thin wall needle resides inside the vein. After confirmation of venous placement, wire threading is allowed to take place. If there is any uncertainty that the catheter or wire resides inside the vein, confirmation is required using ultrasound, TEE, continuous ECG or fluoroscopy. After venous residence of the wire has been confirmed, insertion of a dilator or large bore catheter may then proceed2.

After catheterization and before use, catheter residence in the venous system should again be confirmed as soon as clinically appropriate via either manometry, pressure-waveform measurement or contrast-enhanced ultrasound2. In our first and second case, the decision to confirm the suspected intraarterial placement of the catheter was hastened by the fact that spontaneousfilling of the three lumens with pulsating blood was observed, a finding that suggests intraarterial placement being probable. Intraarterial placement of the CVC was confirmed in both cases by the use of pressure-waveform measurement.

<<In case of a confirmed accidental intra-arterial cannulation a number of promising therapeutic approaches are available according to the literature. Current ASA guidelines2 state that, when such an inadvertent cannulation of an arterial vessel occurs, the catheter should be left in place and immediate consultation with a general surgeon, a vascular surgeon or an interventional radiologistshould be sought to construct an appropriate plan for catheter removal. Treatment should be guided based on existing resources and expertise and should be individualized based on a number of factors (e.g. patient comorbidities, anticoagulation status, CVC diameter, site of injury, time taken for injury to be recognized etc.)2.

Treatment options include open surgical exploration and vascular repair10,13, percutaneous closure devices9,14, or -as in our case- temporary balloon tamponade with concurrent external manual pressure application16 or external manual compression in a palpable artery like the common carotid, where balloon tamponade is best avoided.No specific guidelines or high-level evidence exist for the proper management of these patients, and decisions are best individualized. However,based on literature reviews,a number of treatment algorithms have been proposed17,18.Open surgery seems to be the optimum treatment, however endovascular treatments seem to be only marginally less successful (94.6% for successful endovascular repair vs 100% for open repair), and treatment should be individualized as endovascular treatment offers advantagesover open surgery in some cases18. Endovascular repair might be an attractive option for patients with significant comorbidities or with acute illness, that might be benefited by avoiding general anesthesia.Endovascular techniques shouldalso especially be considered for injuries to the subclavian artery that is not only difficult to compress but also difficult to access surgically17,18.Open surgical management is recommended when the injury is recognized more than 4 hour after cannulation, or when there is no endovascular treatment service18.>>
